# Combined Application of Pan-AKT Inhibitor MK-2206 and BCL-2 Antagonist Venetoclax in B-Cell Precursor Acute Lymphoblastic Leukemia

**DOI:** 10.3390/ijms22052771

**Published:** 2021-03-09

**Authors:** Anna Richter, Elisabeth Fischer, Clemens Holz, Julia Schulze, Sandra Lange, Anett Sekora, Gudrun Knuebel, Larissa Henze, Catrin Roolf, Hugo Murua Escobar, Christian Junghanss

**Affiliations:** Department of Medicine, Clinic III—Hematology, Oncology, Palliative Medicine, Rostock University Medical Center, 18057 Rostock, Germany; elisabeth.fischer2@uni-rostock.de (E.F.); Clemens.holz@uni-rostock.de (C.H.); julsch93vb@hotmail.de (J.S.); Sandra.lange@med.uni-rostock.de (S.L.); anett.sekora@med.uni-rostock.de (A.S.); Gudrun.knuebel@med.uni-rostock.de (G.K.); Larissa.henze@med.uni-rostock.de (L.H.); catrin.roolf@gmail.com (C.R.); hugo.murua.escobar@med.uni-rostock.de (H.M.E.); christian.junghanss@med.uni-rostock.de (C.J.)

**Keywords:** AKT inhibition, MK-2206, acute lymphoblastic leukemia, venetoclax, apoptosis

## Abstract

Aberrant PI3K/AKT signaling is a hallmark of acute B-lymphoblastic leukemia (B-ALL) resulting in increased tumor cell proliferation and apoptosis deficiency. While previous AKT inhibitors struggled with selectivity, MK-2206 promises meticulous pan-AKT targeting with proven anti-tumor activity. We herein, characterize the effect of MK-2206 on B-ALL cell lines and primary samples and investigate potential synergistic effects with BCL-2 inhibitor venetoclax to overcome limitations in apoptosis induction. MK-2206 incubation reduced AKT phosphorylation and influenced downstream signaling activity. Interestingly, after MK-2206 mono application tumor cell proliferation and metabolic activity were diminished significantly independently of basal AKT phosphorylation. Morphological changes but no induction of apoptosis was detected in the observed cell lines. In contrast, primary samples cultivated in a protective microenvironment showed a decrease in vital cells. Combined MK-2206 and venetoclax incubation resulted in partially synergistic anti-proliferative effects independently of application sequence in SEM and RS4;11 cell lines. Venetoclax-mediated apoptosis was not intensified by addition of MK-2206. Functional assessment of BCL-2 inhibition via Bax translocation assay revealed slightly increased pro-apoptotic signaling after combined MK-2206 and venetoclax incubation. In summary, we demonstrate that the pan-AKT inhibitor MK-2206 potently blocks B-ALL cell proliferation and for the first time characterize the synergistic effect of combined MK-2206 and venetoclax treatment in B-ALL.

## 1. Introduction

Therapeutic regimens for acute lymphoblastic leukemia (ALL) have improved during the past decades. However, while most pediatric patients recover after therapy adults still face a grim prognosis with only 10–25% surviving for five years after diagnosis [1]. Resistance to routinely used chemotherapeutic agents is frequently observed and results in even worse outcomes in relapsed ALL patients [2]. This demonstrates the urgent need for novel drugs and targeted treatment strategies aiming at the individual resistance-inducing structures. 

In B-cell precursor ALL (B-ALL) a large variety of factors like mutations, chromosomal translocations or epigenetic dysregulation contribute to tumorigenesis [3,4]. Those molecular aberrations result in aberrant proliferation and apoptosis signaling, ultimately leading to increased tumor cell growth and survival. The PI3K/AKT pathway is a key signaling cascade involved in both processes and frequently dysregulated in several hematological entities including B-ALL [5,6,7]. It further mediates resistance towards chemotherapeutic agents like glucocorticoids and poor prognosis in B-ALL [8]. Typical PI3K/AKT pathway alterations leading to leukemogenesis are mutations in tyrosine kinase genes (*KIT, FMS, FLT3*) or downstream signaling members like PI3K isoenzymes, *PTEN* or Ras oncoproteins [9]. Changes in protein expression and activity of PTEN, CK2 and AKT further result in increased tumor cell proliferation [10]. Aberrant PI3K/AKT signaling ultimately leads to uncontrolled cell growth and blockade of apoptotic cascades via several downstream proteins like GSK3β or 4EBP1 [9].

The kinase AKT is one of the main molecules within the PI3K/AKT pathway and often upregulated in hematologic neoplasms [8,11]. Therefore, pharmacological inhibition of AKT is a promising approach for anti-leukemia treatment. In B-ALL several PI3K/AKT pathway inhibitors have been designed and evaluated both preclinically and clinically with most studies focusing on PI3K and mTOR complexes [10]. Development of selective AKT inhibitors has been difficult due to structural similarities with related kinases [12]. However, due to its high oncogenic potential and frequent dysregulation it remains a key target for pharmacological intervention. Regarding B-ALL only few groups investigated the effect of sole AKT inhibitors: Levy et al. demonstrated that GSK690693 acted anti-proliferative and induced apoptosis [13,14] while three other manuscripts investigated pan-AKT inhibitor MK-2206 [15,16,17]. Both drugs induced significant anti-proliferative effects in leukemic cell lines and changes in AKT downstream protein phosphorylation.

MK-2206 is a potent, selective and orally available small molecule currently investigated in several clinical trials for solid tumors [clinicaltrials.gov]. The initial results are promising with anti-tumor activity (complete or partial response) reported for breast cancer [18,19,20,21], gastrointestinal cancers [20,21] and other entities after mono or combination therapy [22,23]. While MK-2206 mono application had limited anti-tumor activity in hematologic neoplasms [24,25] combination strategies seemed promising. In relapsed chronic lymphoblastic leukemia patients an overall response rate of 92% was achieved with MK-2206 in combination with bendamustine and rituximab [26], raising hopes for further investigation of other arrangements. Several preclinical studies have been conducted for T-cell ALL [27,28,29,30], demonstrating broad anti-proliferative activity, but data on B-ALL is sparse, especially for primary samples and combinatorial approaches [15,16,17]. 

Leukemic cells often lack the inhibition of intrinsic apoptosis. Increased anti-apoptotic BCL-2 signaling is frequently observed in ALL cells as well as further hematological neoplasms [31]. BCL-2 downstream target and pathway member BAD directly interacts with AKT during apoptosis induction [32], justifying mutual targeting of AKT and BCL-2. We therefore aim to characterize the effect of MK-2206 alone as well as in combination with BCL-2 inhibitor venetoclax on B-ALL cell lines and primary samples in a protective co-culture environment. The investigation of this previously untested combination might offer insights into a potentially synergistic mechanism for acute leukemia abrogation.

## 2. Results

### 2.1. MK-2206 Influences AKT and PI3K Signaling Pathway Activity

Previous studies evaluating MK-2206 anti-tumor efficacy in ALL mainly focused on the absolute effect of the substance, but dose-dependencies and duration of the AKT inhibition remained uninvestigated. We thus incubated B-ALL cell lines SEM, RS4;11, REH and NALM-6 with increasing concentrations of MK-2206 for 0.5 h to 72 h. AKT activity was measured by AKT phosphorylation. Basal pAKT expression was high in SEM and RS4;11 cells, while levels were lower in NALM-6 and very weak in REH cells (Figure 1). Incubation with MK-2206 induced an initial decrease in AKT activity in all cell lines which was detectable after 30 min incubation already, even at concentrations as low as 0.5 µM. Although the highest concentration (5 µM) resulted in the strongest dephosphorylation, lower concentrations induced comparable effects. After extended MK-2206 exposure for up to 72 h, AKT phosphorylation remained at a low level in SEM cells. RS4;11 and NALM-6 cells on the other hand reached a dephosphorylation peak after 4 h before returning to initial pAKT levels after further MK-2206 incubation. Due to the very low basal pAKT expression, changes in REH are difficult to observe and interpret but suggest a decrease in AKT phosphorylation. Total AKT expression remained constant throughout the entire observation period. 

We next evaluated the effect of MK-2206 on PI3K/AKT pathway regulation and related signaling proteins (Appendix A). Protein expression and phosphorylation of AKT downstream target 4EBP1 was analyzed by immunoblot and demonstrated reduced protein phosphorylation, thus indicating increased 4EBP1 activity after MK-2206 treatment. Dephosphorylation of 4EBP1 was observed in all cell lines starting as early as 30 min MK-2206 incubation period, while a decrease in total 4EBP1 expression was also detected in RS4;11 and NALM-6 at later time points. Further, ERK1/2 was analyzed to investigate whether MK-2206-mediated AKT inhibition had an influence on MAPK signaling. Indeed, a concentration-dependent ERK dephosphorylation was observed in SEM and RS4;11 cells after extended incubation periods. Total ERK levels were not affected. Similar to MK-2206-induced AKT dephosphorylation PI3K/AKT signaling regulation was rather low in REH cells.

### 2.2. MK-2206 Acts Anti-Proliferative Independently of Apoptotic Signaling

To evaluate whether the observed AKT dephosphorylation is translated into regulation of proliferation or metabolism, we performed two assays to assess anti-proliferative or anti-metabolic effects after MK-2206 incubation (Figure 2). As expected, cell lines SEM, RS4;11 and NALM-6 which demonstrated broad AKT dephosphorylation showed a concentration-dependent decline in cell proliferation and metabolic activity after both 48 h and 72 h incubation. Interestingly, also REH cells exhibiting a very low baseline AKT activity responded well to MK-2206 treatment. Incubation with 0.05 µM MK-2206 for 72 h resulted in a significant reduction of metabolic activity. Significant anti-proliferative effects were obtained using 0.1 µM MK-2206.

To further investigate the mechanistic origin of the observed anti-proliferative effects we next performed microscopic evaluation of the cell morphology (Figure 3a). With increasing MK-2206 concentrations all cell lines showed signs of disintegrating cell membranes. SEM and RS4;11 cells exhibited strong vacuolization, increasing amounts of cellular debris and nuclear fragmentation with stable nuclear-to-cytoplasm ratio. Cytoplasmatic blebs were further detected in RS4;11 cells. Cell shrinkage including karyopyknosis and chromatin condensation was observed in NALM-6 and REH cells while the nuclear-to-cytoplasm ratio remained constant. 

We next evaluated if those morphological changes are evoked by MK-2206-mediated increased induction of apoptosis. We therefore performed immunoblotting of apoptosis marker Cleaved Caspase-3 (Appendix A). No changes in Cleaved Caspase-3 expression were observed after 0.5 h to 72 h incubation with increasing concentrations of MK-2206 compared to DMSO-treated control cells, indicating apoptosis-independent mechanisms in all four cell lines. To validate this finding we further performed annexin V/propidium iodide (PI) staining and subsequent flow cytometric analysis in SEM and RS4;11 cells (Figure 3b,c). This analysis confirmed our previous results, demonstrating no induction of apoptosis in both cell lines. 

### 2.3. MK-2206 Reduces Cell Viability of Primary Blasts in Protective Microenvironment

After successful evaluation of MK-2206 in B-ALL cell lines we subsequently tested the drug in two primary human B-ALL model systems. After harvesting from xenograft mice, cells were maintained in co-culture with murine bone marrow stromal cells, simulating the protective bone marrow microenvironment. Inhibitor studies were only conducted during primary cell steady proliferation time frames. The two patient samples (#0122, #0159) were selected based on their basal pAKT expression, with patient #0122 exhibiting a modest AKT phosphorylation while patient #0159 showed a strong positive signal for pAKT in both immunofluorescence staining and immunoblot (Figure 4a,b). Both patients further harbor the t(4;11) *KMT2A* translocation and *TP53* Pro72Arg hotspot mutation. A *KRAS* Gly12Ser mutation was further detected in patient #0122. Patient #0159 possesses heterozygous pathogenic mutations in *JAK3* and *MET* as well as a *PIK3CA* Gln546Lys mutation with low allele frequency (Figure 4c).

Both co-culture systems were incubated with increasing concentrations of MK-2206 for 72 h and blast viability was assessed (Figure 4d). Interestingly, cell vitality dropped independently of pAKT expression with significant reductions starting at 1.0 µM and 0.5 µM for patients #0122 and #0159, respectively. Comparable results were achieved with annexin V/PI apoptosis staining: In contrast to cell line experiments apoptosis was induced in both primary cell cultures (Figure 4e). After incubation with 0.5 µM and 5.0 µM MK-2206 the amount of viable cells dropped from 72.3% to 53.9% and 27.4% for patient #0122, and from 86.6% to 70.5% and 42.4% for patient #0159, respectively.

**Figure 4 ijms-22-02771-f004:**
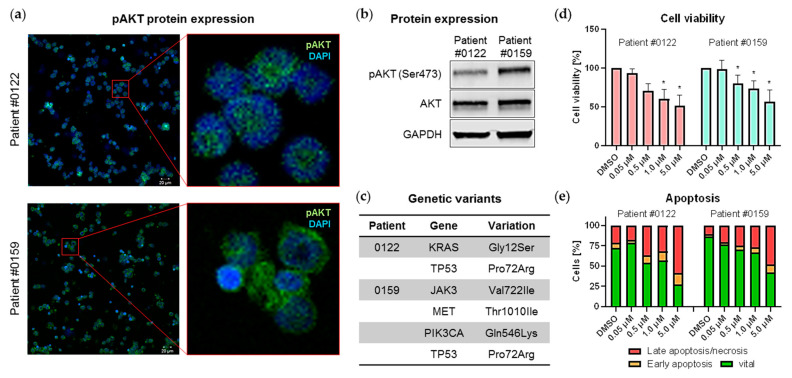
Effects of MK-2206 on primary B-ALL blasts. (**a,b**) Protein expression of phosphorylated and total AKT was measured by immunofluorescence staining (**a**) and immunoblotting (**b**). Cells of patients #0122 and #0159 were orthotopically xenografted into NSG mice as described in [33,34] and spleen cells were subsequently harvested for further analysis. (**a**) Expression of pAKT was determined by immunofluorescence using the LSM780 confocal microscope (Zeiss) at 20-fold magnification (left panel). The right hand images are enlargements of the regions indicated in the left hand red boxes. (**b**) Cells were lysed and analyzed for phosphorylated and total AKT protein expression by immunoblot. GAPDH was used as internal loading control. Blots were processed and cropped using Image Studio Lite 5.2 software and MS PowerPoint (2011) to improve clarity and conciseness. (**c**) Genetic variants in both patients were detected using Cancer hotspot panel (Ion PGM System, Thermo Fisher Scientific) and previously described in [34]. (**d,e**) Xenograft-derived primary blasts were cultured on a murine stromal cell feeder layer system and cell viability and apoptosis induction were assessed after 72 h incubation with increasing concentrations of MK-2206. (**d**) Cell viability was determined by CellTiter Blue assay in technical triplicates. Mean ± standard deviation of three individual biological replicates. Significance was determined by student’s t test after testing Gaussian distribution. * *P* < 0.05. (**e**) Cells were stained with anti-human CD19, annexin V-FITC and PI for apoptosis analysis by flow cytometry. Cells positive for annexin V and negative for PI are indicated as early apoptotic whereas cells positive for both markers are considered late apoptotic/necrotic. Human primary blasts and murine feeder cells were discriminated by anti-human CD19 staining.

### 2.4. Addition of BCL-2 Inhibitor Venetoclax Synergistically Reduces Tumor Cell Proliferation

Evasion of apoptosis is a key feature of successful tumor cells. As MK-2206 alone was not able to induce apoptosis (Figure 3b) in the cell lines tested earlier we subsequently tested MK-2206 in combination with BCL-2 inhibitor venetoclax. Cell lines SEM and RS4;11 were selected based on their rather high basal pAKT expression as well as their molecular subtype, sharing the *KMT2A* rearrangement with the tested primary samples. IC20/30 concentrations for combinatory experiments were selected to allow for determination of synergistic effects. We first evaluated whether either substance alone or in combination induced cytotoxic effects on healthy cells. Neither MK-2206 or venetoclax alone nor the combination resulted in hemolysis (Figure 5a) or decreased PBMC viability (Figure 5b). We then incubated SEM and RS4;11 cells with MK-2206 and venetoclax simultaneously for 72 h and measured cell proliferation and metabolic activity (Figure 5c,d). Compared to untreated control cells the combination induced a significant reduction of cell numbers and cellular metabolism in both cell lines. Calculation of Bliss values (Figure 5f) indicated synergistic anti-metabolic effects in SEM cells while no synergy was detected in RS4;11. Mild synergistic anti-proliferative responses were present in both cell lines (SEM: 0.0452; RS4;11: 0.0820).

**Figure 5 ijms-22-02771-f005:**
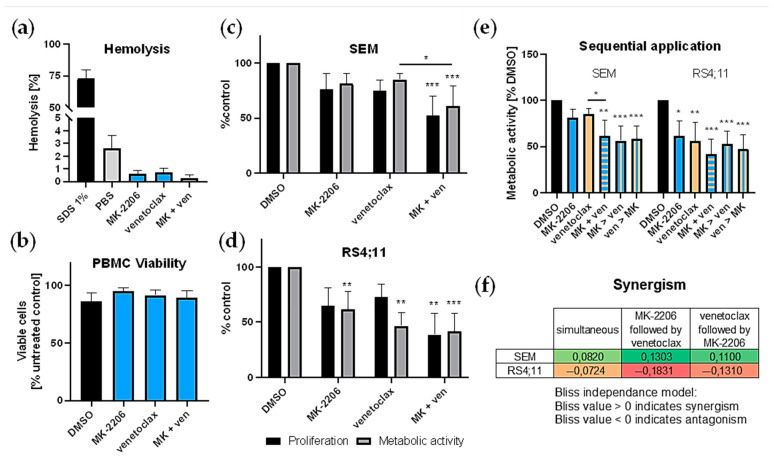
Combined application of MK-2206 (MK) and venetoclax (ven). (**a**,**b**) Cytotoxicity of MK-2206 (0.25 µM), venetoclax (10 nM) or the combination was determined by hemolysis assay and PBMC viability analysis. Blood of five healthy voluntary donors was used. Technical triplicates were performed for both assays. Mean ± standard deviation. (**a**) Hemolytic activity was assessed by hemoglobin release after 120 min incubation with the substances, 1% SDS (positive control) or PBS (negative control). (**b**) PBMCs were isolated by density gradient centrifugation and incubated with the respective inhibitors or 10 µM DMSO (control) for 24 h. Cell viability was determined by Calcein AM assay. (**c,d**) Assessment of cell proliferation and metabolic activity was performed by trypan blue staining and WST-1 assay, respectively and compared to DMSO-treated control cells. Cells were incubated with 0.25 µM MK-2206, 10 nM (SEM (**c**)) or 2.5 nM (RS4;11 (**d**)) venetoclax or both inhibitors for 72 h. (**e**) Effects of sequential application of MK-2206 and venetoclax on SEM and RS4;11 cells were determined by WST-1 assay. Substances were applied either simultaneously (MK + ven) or the second substance was added 24 h after the first (MK > ven or ven > MK). Metabolic activity was assessed after 72 h. Mean ± standard deviation of at least three individual biological replicates. Significance was determined by one-way ANOVA. * *P* < 0.05; ** *P* < 0.01; *** *P* < 0.001. Asterisks demonstrate significance compared to DMSO-treated control cells or other treatments when connected by lines. (**f**) Synergy was calculated according to the Bliss independence model with values > 0 indicating synergistic combinations [35]. Bliss values were calculated based on the mean biological response of at least four biological replicates. Cells were incubated with MK-2206, venetoclax or both (simultaneously or sequentially) and metabolic activity was assessed by WST-1 assay after 72 h.

The sequence of drug application can be crucial for the success of combinatory approaches. We thus tested MK-2206 and venetoclax simultaneous incubation for 72 h and compared the effect to cells, where one agent was applied first and the other was added 24 h later (Figure 5e). In SEM cells synergistic effects were obtained with all application sequences. The best results were achieved when MK-2206 was applied first and venetoclax was added second. However, this application schedule was not significantly superior to the other regimens. Interestingly, MK-2206 treatment followed by venetoclax was the weakest combination in RS4;11 cells with a Bliss value well below zero (−0.1831), indicating antagonistic effects. When venetoclax was given first and MK-2206 was added later or when both drugs were applied simultaneously, effects were slightly stronger but still not synergistic.

### 2.5. Combined MK-2206 and Venetoclax Application Increases Bax Translocation

To evaluate whether addition of venetoclax could synergistically induce apoptosis we then performed annexin V/PI staining. As expected already very low concentrations of BCL-2 inhibitor venetoclax (SEM: 10 nM, RS4;11: 2.5 nM) reduced the number of viable blasts and increased the amount of early and late apoptotic cells (Figure 6a). Apoptosis induction was significantly increased in SEM cells incubated with both, venetoclax alone and in combination with MK-2206 compared to untreated controls. In contrast to the experiments demonstrating synergistic anti-proliferative activity, no such effect was observed analyzing apoptosis induction. Compared to venetoclax mono therapy the amount of viable and apoptotic cells did not change when MK-2206 was added. RS4;11 cells resembled the same pattern.

We also performed a functional assay to evaluate the influence of both drugs on the BCL-2 signaling pathway-mediated apoptosis cascade. During apoptosis induction the pro-apoptotic protein Bax, which is blocked by active BCL-2, translocates from cytoplasm to the mitochondrial wall. Pores are created by Bax dimerization resulting in cytochrome c release and subsequent loss of membrane potential and apoptosis commitment. The immunofluorescence imaging-based Bax translocation assay detects spatial overlap between Bax and mitochondria, thus demonstrating functional effects of upstream BCL-2 inhibition on apoptosis (Figure 6c). Pearson’s coefficients were calculated from the obtained images to quantify the amount of Bax protein located at mitochondria (Figure 6b). Although not significant, a slight increase in Bax/mitochondria overlap was observed after venetoclax mono therapy as well as combined MK-2206 and venetoclax incubation in SEM cells. Single MK-2206 application did not result in increased Bax translocation. The Pearson’s coefficient mean remained at the same level after MK-2206 or venetoclax mono therapy in RS4;11 cells. Similar to SEM cells, combined application resulted in a minor rise of Bax/mitochondria overlap.

## 3. Discussion

AKT is a kinase involved in numerous regulatory mechanisms affecting cell growth, proliferation, metabolism and survival. Dysregulated PI3K/AKT signaling is observed in many cancer entities including hematologic malignancies, suggesting AKT as a possible target for anti-leukemic targeting [36]. Several AKT inhibitors have been pre-clinically tested to induce anti-tumorigenic effects in B-ALL [10,37]. We herein evaluated the effects of the selective pan–AKT inhibitor MK-2206 on B-ALL cell lines and primary blasts cultivated in a protective bone marrow microenvironment. 

Other groups have previously demonstrated successful AKT inhibition in leukemia cell lines [13,30,38,39]. Matching our data Simioni et al. detected decreased AKT phosphorylation in SEM cells incubated with 0.5 µM MK-2206 for 30 min [15]. Han et al. obtained similar results in NALM-6 cells [16] and Naderali et al. observed AKT dephosphorylation in NALM-6 and REH cells [17]. However, both studies only surveyed one particular time point using a single concentration of MK-2206. We herein demonstrated a concentration-dependent reduction of AKT phosphorylation in B-ALL cell lines for the first time. In addition, the observed cell lines reacted differently after initial pAKT diminishment. While phosphorylation levels remained low in SEM and RS4;11, also for extended incubation periods, AKT activity increased rapidly and matched the basal setting after 24 h in NALM-6 cells. This indicates a possible feedback mechanism in NALM-6 which might be induced by increased PI3K upstream signaling, resulting in AKT phosphorylation and thus activation. Drug evasion regulation loops are commonly observed after treatment with selective kinase inhibitors. In a previous study, we detected increased AKT phosphorylation after prolonged incubation with CX-4945, an inhibitor targeting AKT regulating kinase CK2 [40]. In addition to AKT, translation controlling factor 4EBP1 was also dephosphorylated, indicating functional MK-2206-induced regulation of downstream proteins. This matches the data of other studies investigating the influence of MK-2206 on B-ALL cell lines demonstrating PI3K/AKT signaling modulation [5,15,16]. Those effects were more prominent in SEM and RS4;11 cells, possibly due to higher basal AKT phosphorylation.

As AKT regulation after prolonged MK-2206 incubation was strongest and persistent in SEM and RS4;11 cells, we expected higher anti-proliferative effects in those cell lines. Interestingly, proliferation and metabolic activity of all cell lines were reduced independently of their basal AKT activity and duration of dephosphorylation. This phenomenon suggests that MK-2206 induced off-target effects by binding a structurally related molecule so far unidentified. This seems probable as the development of selective AKT inhibitors has long been hampered due to chemical similarities with other kinases [12]. Neri et al., Simioni et al. and Naderali et al. also tested cell viability of B-ALL cell lines after MK-2206 incubation with increasing concentrations [5,15,17]. IC50 values were slightly higher than in our study, which might be explained by shorter incubation periods (48 h vs. 72 h) and different readout assays (MTT vs WST-1 assay). Gorlick et al. on the other hand determined similar values in NALM-6 cells but RS4;11 cells were more sensitive [41]. Importantly, significant anti-leukemic results were obtained with physiologically achievable concentrations in all cell lines. 

In addition to cell lines we also evaluated the effect of MK-2206 on primary B-ALL blasts in a protective bone marrow feeder cell co-culture system. To our knowledge this is the first study assaying the anti-leukemic activity of MK-2206 on primary B-ALL cells in a co-culture microenvironment. The inhibitor induced significant anti-proliferative effects in both primary samples tested. Notably, the decrease in cell vitality was again not associated with basal pAKT expression, underlining the results discussed earlier. The presence of a *PIK3CA* mutation as well as other pathogenic hotspot variants in patient #0159 did not interfere with the induction of MK-2206-induced effects as well. The fact that the inhibitor produced significant anti-leukemic effects in primary samples harboring *KMT2A* rearrangements, which are associated with a very grim prognosis, further underlines its potential. Neri et al. used four different B-ALL patient samples in monoculture and observed a similar reduction of cell viability at comparable concentrations [5]. Two of the four patients had high basal AKT phosphorylation while data was not shown for the other two. Further studies confirmed anti-leukemic effects in primary T-ALL cells both in mono and co-culture [27,28]. 

Interestingly, and in contrast to the evaluated primary samples, MK-2206 did not induce apoptosis in any of the observed cell lines. This might be due to the technical fact that primary cultures are more fragile and vulnerable than robust and immortal cell culture systems. Otherwise, the molecular background of the patient samples is different from the cell lines. We thus questioned if a combined application of MK-2206 and venetoclax, an inhibitor of anti-apoptotic signaling molecule BCL-2, could increase the anti-leukemic effects of the AKT inhibitor. As this combination is tested in acute leukemia cell lines for the first time and was not evaluated regarding cytotoxic effects on healthy blood cells, we first demonstrated that the application was safe and not affecting erythrocytes and PBMCs after short-term incubation. On the other hand, proliferation and metabolic activity of tumor cells were reduced compared to the mono substances and synergistic effects were calculated for SEM cells independently of the application sequence. This matches the data of Shen et al. who evaluated this combination in primary chronic lymphocytic leukemia cells [42]. To check whether the increased cytostatic effect was due to raised apoptosis signaling we performed functional apoptosis analysis on a molecular level. Matching existing literature we demonstrated elevated apoptosis levels after venetoclax incubation [43,44,45], and also following combined application by annexin V staining for flow cytometry. Venetoclax mono therapy, however, was just as effective as the combination. This contrasts with the findings of a previous study in diffuse large B-cell lymphoma cell line RC and mantle cell lymphoma cell line MINO demonstrating synergistic induction of apoptosis [46]. This study used a 40-fold higher concentration of MK-2206 compared to our experiments though, possibly explaining the discrepancy. We subsequently employed the Bax translocation assay to assess whether the observed apoptosis was induced by inhibited BCL-2 signaling. With the low concentrations used we did not detect any relevant elevated Bax translocation towards mitochondria after MK-2206 or venetoclax monoapplication. A mild trend towards increased overlap of both structures after combined application of both drugs suggested mild additive or synergistic BCL-2 pathway-mediated anti-apoptotic signaling. It remains to be evaluated why those effects are not detected using annexin V staining. Assay sensitivity, effect strength and analysis time frames might play a role for this question.

In summary, this study investigates the short-term mechanistic effects of MK-2206 incubation, demonstrating that MK-2206 application is a powerful tool to induce a significant decrease in leukemia proliferation throughout observation periods up to 72 h. This was true even in primary blasts cultivated in a protective microenvironment. It underlines the anti-leukemic potential of AKT inhibition and evaluates the synergistic combined application with venetoclax previously untested in ALL. These results raise hopes for further long-term observation and in vivo evaluation of the AKT inhibitor alone or in combination with other small molecules or conventional chemotherapy.

## 4. Materials and Methods

### 4.1. Inhibitors, Cell Lines and Cell Culture

MK-2206 was obtained from Selleckchem (Houston, TX, USA) and venetoclax was purchased from MedChemExpress (Monmouth Junction, NJ, USA). 

Human B-ALL precursor cell lines SEM, RS4;11, NALM-6 and REH were purchased from German Collection of Microorganisms and Cell Cultures (DSMZ, Braunschweig, Germany) and cultured as recommended by the manufacturer. SEM was maintained in IMDM medium, RS4;11 in Alpha MEM medium and NALM-6 and REH in RPMI 1640 medium (all PAN–biotech, Aidenbach, Germany). Media were supplemented with 10% heat-inactivated fetal calf serum (PAN-biotech) and 100 μg/mL penicillin and streptomycin (PAN-biotech) and cells were cultured at 37 °C and 5% CO_2_. Medium was changed twice weekly. After passaging and at the beginning of all experiments cells were seeded at a density of 3.3 × 10^5^ cells per mL medium. Cell lines were regularly checked for authenticity and mycoplasma contamination.

### 4.2. Primary Samples and Ethics

All experiments were performed in accordance to the Declaration of Helsinki and the local ethics committee standards. Mononuclear cells from bone marrow aspirates of two newly diagnosed B-ALL patients (Rostock University Medical Center) were isolated using Biocoll separating solution (Merck Millipore, Darmstadt, Germany) and previously characterized on molecular level using next-generation sequencing (Cancer hotspot panel, Ion PGM System, Thermo Fisher Scientific, Schwerte, Germany) according to the manufacturer’s protocols [34].

### 4.3. Generation of Patient-Derived Xenograft Models

For patient-derived xenograft (PDX) model generation 2.5 × 10^6^ cells per primary sample were orthotopically xenografted into 8–12 week old male or female NOD scid gamma mice (NOD.Cg-*Prkcd^scid^ Il2rg^tm1Wjl^*/SzJ, NSG, Charles River Laboratories, Sulzfeld, Germany). Tumor cell engraftment was regularly evaluated by peripheral blood flow cytometry (anti-human CD45-FITC (clone 2D1) and anti-human CD19-PE (clone 4G7)) measured using FACSVerse and FACSuite software, all Becton Dickinson, Heidelberg, Germany). Mice were euthanized by cervical dislocation when a blast frequency of 30% in peripheral blood was achieved and then bone marrow and spleen cells were isolated. Tumor cells were then serially transplanted into the next PDX generation until generation 3. Mice were bred and housed under specific pathogen-free conditions with access to water and standard chow ad libitum. All experiments were carried out in a laboratory setting and no intervention was performed within the animal housing and breeding rooms. Experiments were approved by the review board of the federal state Mecklenburg-Vorpommern, Germany (reference number: LALLF MV/7221.3–1.1-002/15).

### 4.4. Co-Culture of PDX-Derived Blasts and OP-9 Feeder Cells

PDX-derived primary tumor cells isolated from spleens of the third xenograft generation were subsequently cultured in vitro using murine bone marrow stroma cell line OP-9 as feeder layer. OP-9 was obtained from ATCC (Manassas, VA, USA) and maintained in Alpha medium without ribonucleosides and desoxyribonucleosides (Biochrom, Berlin, Germany) supplemented with 20% fetal calf serum (Biochrom) and 100 μg/mL penicillin and streptomycin (Biochrom). After passaging OP-9 cells were seeded at a density of 2.25 × 10^4^ cells per mL medium. Three days prior to co-culture initiation 2 × 10^5^ OP-9 cells were seeded into a 6 well plate and incubated at 37 °C and 5% CO_2_. OP-9 cells were then irradiated with 15 Gy and fresh medium was added. Afterwards 2–4 × 10^6^ PDX-derived primary tumor cells were added to every well. Medium was changed every two to three days and blasts were transferred to fresh OP-9 feeder cells once per week. Inhibitor experiments only commenced once primary cells demonstrated stable proliferation.

### 4.5. Immunoblot

Cells were lysed using RIPA buffer (Cell Signaling, Danvers, MA, USA) and ultra sound exposure. Proteins were separated on Mini or Midi gels (Bio-Rad, Munich, Germany), blotted onto a PVDF membrane (Bio-Rad) using the Trans-Blot^®^ Turbo™ Transfer System (Bio-Rad, 2.5 A, 25 V, 10 min) blocked in LI-COR (Lincoln, NE, USA) blocking buffer for 1 h and detected via LI-COR Odyssey Imaging System and Image Studio Lite 5.2 software. Antibodies and respective dilutions are listed in Appendix A. Primary antibodies were incubated for 1 h at room temperature or overnight at 4 °C. Secondary antibodies were incubated for 1 h at room temperature. For quantification, band intensities were assessed using Image Studio Lite 5.2 software and the ratio between phosphorylated and corresponding total protein band intensity was calculated.

### 4.6. Cell viability and Vitality Assays

Proliferation was assessed by counting viable cells using trypan blue dye exclusion. Metabolic activity was evaluated by WST-1 assay (Roche, Mannheim, Germany). The CellTiter Blue^®^ (Promega, Madison, WI, USA) assay was used to assess cell viability of primary cells cultured with OP-9 feeder cells. Both assays were carried out in technical triplicates. At least three biological replicates were performed for all analyses.

### 4.7. Cytotoxicity Assays

Blood of five healthy voluntary donors was collected to analyze cytotoxic effects of single or combined inhibitor incubation on erythrocytes and peripheral blood mononuclear cells (PBMC) as previously described [47,48]. Briefly, 20 µL full blood per well was seeded into a 96-well plate and incubated with the indicated inhibitor concentrations, PBS (negative control) or 1% SDS (Merck, Darmstadt, Germany) as a positive control for hemolysis for 120 min. Absorption of the cell-free supernatant was determined at 540 nm (reference 690 nm). To assess the effect on PBMC viability those cells were isolated by density gradient centrifugation using PANcoll (PAN-biotech). 5 × 10^4^ cells per well were seeded into a 96-well plate and incubated with the inhibitors or DMSO for 24 h. After inhibitor incubation 50 µL Calcein AM working solution (final concentration 0.5 µM) per well were added and incubated for 30 min before detection using the Glomaxx plate reader (Promega, excitation 485 nm, emission 535 nm). All samples were measured in technical triplicates.

### 4.8. Pappenheim Staining and Morphology Analysis

To assess morphological changes after inhibitor incubation 5 × 10^4^ cells were spun onto a microscopic slide (10 min, 700 rpm) using the Cytospin 3 centrifuge (Shandon, Frankfurt/Main, Germany). Cytospins were air-dried and stored protected from light until staining, as previously described [49]. In brief, cells were first incubated in May-Grünwald solution (Merck), washed in tap water and subsequently stained with Giemsa solution (Merck) for 20 min. Slides were analyzed using the EVOS xl core microscope at 100-fold magnification. 

### 4.9. Analysis of Apoptosis by Flow Cytometry

Apoptosis was analyzed by annexin V/PI staining and subsequent flow cytometry. After inhibitor incubation cells were harvested and washed twice with PBS. Cells were resuspended in binding buffer, stained with annexin V-FITC (Becton Dickinson) and incubated at room temperature protected from light for 15 min. Propidium iodide (0.6 µg/mL) was added just before analysis (FACSCalibur or FACSVerse, Becton Dickinson) and data was analyzed using CellQuest or FACSuite software (both Becton Dickinson).

### 4.10. Immunofluorescence Imaging

For immunofluorescence-based detection of protein expression and localization 5 × 10^4^ cells were spun onto a microscopic slide (10 min, 700 rpm) using the Cytospin 3 centrifuge (Shandon). Cytospins were air-dried, fixed in ice-cold methanol for 10 min and permeabilized with 50 µL 0.1% Triton X-100 (Sigma-Aldrich, St. Louis, MO, USA) for 10 min at room temperature. Slides were blocked in 1% BSA (Serva, Heidelberg, Germany) for 1 h before incubation with primary antibodies (pAKT Ser473 clone 587F11, 1:50 dilution, Cell Signaling or Bax clone D2E11, 1:50 dilution, Cell Signaling) overnight. The Alexa Fluor 488 goat anti-rabbit antibody (1:1000 dilution, Invitrogen, Carlsbad, CA, USA) was used as secondary antibody and applied for 1 h. Mounting medium containing DAPI (Roti-Mount FluorCare DAPI, Roth, Karlsruhe, Germany) was used for pAKT co-staining while DAPI-free mounting medium was used for Bax analysis (Roti-Mount FluorCare, Roth). To visualize mitochondria MitoSpy™ Red (Biolegend, San Diego, CA, USA) was used. Cells were incubated with 250 nM MitoSpy™ for 30 min before being harvested and spun onto microscopic slides. Images were taken using the LSM780 confocal microscope (Zeiss, Oberkochen, Germany) or Eclipse TE200 microscope (Nikon, Minato, Japan) and ZEN Imaging software (Zeiss).

### 4.11. Colocalization Analysis Using Bax Translocation Assay

During apoptosis the pro-apoptotic protein Bax shuffles from cytoplasma to the mitochondrial wall and induces pore-like structures resulting in cytochrome c release and cell death. By immunofluorescence co-staining of Bax and mitochondria (see above) it is possible to estimate the amount of Bax protein localized at mitochondria as a parameter for functional BCL-2 pathway-related apoptosis induction. While Bax is displayed in green and mitochondria are stained in red a yellow signal indicates regions with green and red signal overlap, suggesting colocalization of Bax and mitochondria. Color-merged images were split into separate images for both channels and regions of interest were marked using Fiji software to exclude noise signal and damaged cells adulterating data analysis. For each image colocalization analysis of Bax and mitochondria was performed using the Coloc2 plugin in Fiji and subsequent Pearson’s coefficient calculation to quantify the Bax/mitochondria overlap.

### 4.12. Statistics

For cell proliferation, vitality and viability assays all tests were performed in at least three individual biological replicates and three technical replicates were analyzed for WST-1 and CellTiter Blue^®^ assays. Mean and standard deviation of at least three individual biological replicates were calculated and significance was determined by student’s t test or ANOVA after testing Gaussian distribution.

Synergy of MK-2206 and venetoclax combination was calculated according to the Bliss independence model. In this test the observed effect of a combinatory treatment is compared to a calculated expected effect. The expected effect (E) is calculated as follows: E = (A + B) − (A × B), where A and B represent the relative inhibition of single agents A and B. The difference between the observed and expected effect (Δ = O − E) of the combinatory treatment then determines the level of synergy. If the observed inhibitory effect is higher than the expected value (Δ > 0) the combination is referred to as synergistic while Δ < 0 is considered antagonistic.

For colocalization analyses a minimum of ten images per treatment group (control, MK-2206, venetoclax, combination) were analyzed (four biological replicates and two to four individual images per biological replicate). Pearson’s coefficients were calculated for each image using the Coloc2 plugin in Fiji and ANOVA was applied to calculate differences between the study groups.

The following levels of significance were determined for all statistical analyses: *p* < 0.05 (*), *p* < 0.01 (**) and *p* < 0.001 (***).

## Figures and Tables

**Figure 1 ijms-22-02771-f001:**
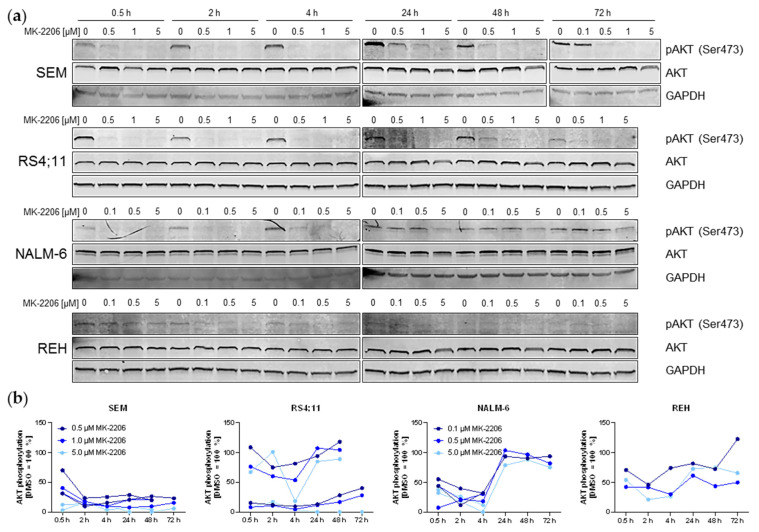
Expression of phosphorylated and total AKT protein determined by immunoblot and subsequent quantification. (**a**) SEM, RS4;11, REH and NALM-6 cells were incubated with the stated concentrations of MK-2206 for the indicated time periods. The displayed immunoblots are representative captions of at least two individual biological replicates. GAPDH was used as internal loading control. Blots were processed and cropped using Image Studio Lite 5.2 software and MS PowerPoint (2011) to improve clarity and conciseness. (**b**) Phospho (p)-AKT and total AKT band intensities of two independent biological replicates including the blots shown in Figure 1a were determined using Image Studio Lite 5.2 software. AKT phosphorylation was calculated as the ratio of pAKT/total AKT and normalized to the DMSO control separately for every time point. One REH replicate was not quantified due to the very low basal pAKT expression and therefore low signal to noise ratio not acceptable for reliable quantification.

**Figure 2 ijms-22-02771-f002:**
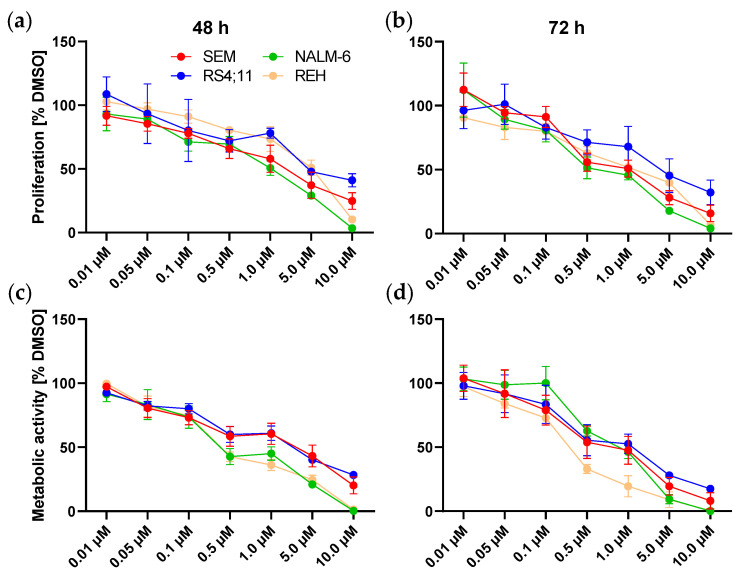
Effects of MK-2206 incubation on cell proliferation and metabolic activity. Dose-response curves of SEM, RS4;11, REH and NALM-6 cells incubated with increasing concentrations of MK-2206 for 48 h (**a**,**c**) and 72 h (**b**,**d**). Proliferation (**a**,**b**) was assessed by trypan blue staining and metabolic activity (**c**,**d**) was determined by WST-1 assay. Mean ± standard deviation of at least three individual biological replicates. Significance was determined by student’s t test after testing Gaussian distribution. To increase clarity, significance levels are not indicated within the graphs and instead listed in Appendix A.

**Figure 3 ijms-22-02771-f003:**
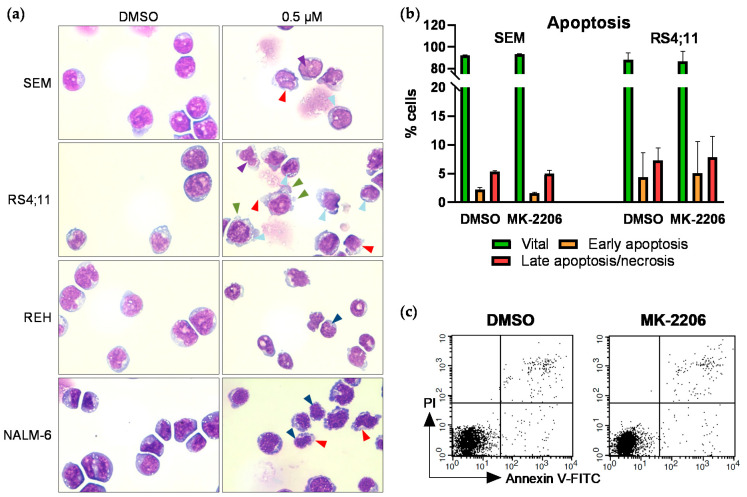
Effects of MK-2206 on cell morphology and apoptosis induction. (**a**) Cells were incubated with increasing concentrations for 72 h. Cytospins were prepared and Pappenheim stained. Representative images of all cell lines treated with 0.5 µM MK-2206, 100-fold magnification. Images were acquired using the EVOS xl core microscope (Nikon). Red arrows: disintegrating cell membrane; light blue arrows: vacuolization; purple arrows: nuclear fragmentation; green arrows: cytoplasmic blebs; dark blue arrows: karyopyknosis. (**b**) SEM and RS4;11 cells were incubated with 0.25 µM MK-2206 for 72 h and subsequently stained with annexin V-FITC and PI for apoptosis analysis by flow cytometry. Cells positive for annexin V and negative for PI are indicated as early apoptotic whereas cells positive for both markers are considered late apoptotic/necrotic. (**c**) Exemplary dot plots of flow cytometric apoptosis determination in SEM cells.

**Figure 6 ijms-22-02771-f006:**
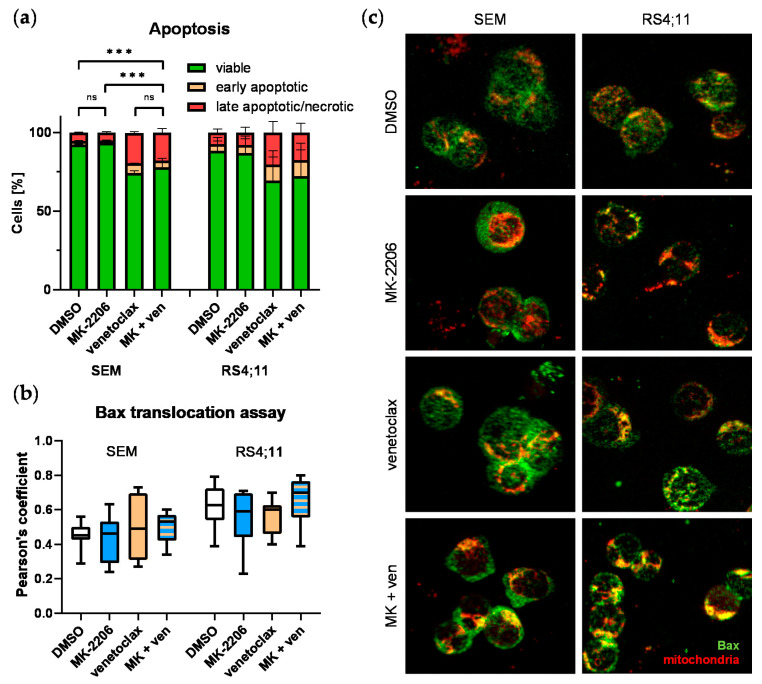
Analysis of apoptotic processes after single or combined MK-2206 (MK; 0.25 µM) and venetoclax (ven; 10 nM (SEM) or 2.5 nM (RS4;11)) incubation in SEM and RS4;11. (**a**) Cells were incubated for 72 h and stained with annexin V-FITC and PI. The amount of apoptotic cells was determined by flow cytometry with cells single positive for annexin V called early aposptotic and cells positive for both markers late apoptotic/necrotic. Mean ± standard deviation of at least three individual biological replicates. Significance was determined by one-way ANOVA. *** *P* < 0.001. (**b,c**) Downstream apoptotic signaling was investigated by Bax translocation assay. Cells were treated with one or both inhibitors for 48 h and spun onto microscopic slides. Mitochondria were stained using MitoSpy™ red and Bax protein was stained in green. Images were taken using the Eclipse TE200 microscope (Nikon) and at 40-fold magnification. Four biological replicates were performed for each treatment and two to four individual images per biological replicate were considered for colocalization analysis. (**b**) Color-merged images were split into separate images for both channels and regions of interest were marked using Fiji program. Colocalization analysis of Bax and mitochondria was performed using the Coloc2 plugin in Fiji and Pearson’s coefficient were used to quantify the amount of Bax/mitochondria overlap. (**c**) Representative images of SEM and RS4;11 cells treated with DMSO (control), MK-2206, venetoclax or both. Mitochondria are stained in red and Bax in green. Yellow signals are a product of red and green signal overlay and indicate colocalization of both structures.

## Data Availability

The datasets used and/or analyzed during the current study are available from the corresponding author on reasonable request.

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
