# Peer review of "Combined Application of Pan-AKT Inhibitor MK-2206 and BCL-2 Antagonist Venetoclax in B-Cell Precursor Acute Lymphoblastic Leukemia"

_ijms, 2021, doi:10.3390/ijms22052771_

Round 1
Reviewer 1 Report
Thanks for giving me the opportunity to review this interesting manuscript. From my point of view it is a good manuscript that gives new information about the pan-AKT 28 inhibitor MK-2206 as a potentially blocker of B-ALL cell viability and for the first time the authors characterize the synergistic effect of combined MK-2206 and venetoclax treatment in B-ALL. I think there is no need to perform new experiments, but the manuscript needs re-writing some parts and improving the presentation of some results. Moreover, conclusions obtained from the results should be taken with caution, the study has its limitations. Here are my comments:
LINE 2
I believe the title is too ambitious, the results should be taken with a lot of caution
LINE 45
Specify more clearly the involvement of PI3K/AKT in leukemia
LINE 47
In B-ALL
LINE 51
In B-ALL
LINE 52
Remove kinase
LINE 56
Explain why not to study GSK690693
LINE 55
and...? which parameters did they study? what results did they obtain?
see line 68
LINE 68
Sparse means there are some articles. Which ones? Please describe. See comment on line 290
What's new in your study compared to the previous ones?
LINE 70
Last paragraph of introduction should refer to the aims of the study and not a summary of results, which are described in the abstract.
LINE 71
From the introduction information, there is no reason to use venetoclax, please explain, at least slightly
FIGURE 1
If there are two biological replicates, fig 1b could be the mean of the two replicates and include standard deviations?
LINE 84
If you are talking about all cell lines, you should say 0,5 uM
LINE 85
I am afraid not always. I think you cannot state this.
LINE 88
Please comment on REH cells
FIGURE S1
It would be interesting to see the plots with the band intensities values, as in Fig1b
LINE 111
Shouldn't this sentence go to the previous paragraph? “REH cells demonstrated a very low overall pAKT 111 expression while total protein levels were comparable to the other cell lines.”
LINE 120
Why did you chose these time points? did you test other time points? NALM-6 cell line at 48h recovers initial activity
LINE 122
“0.05 μM MK-2206” I think this could be deleted, since it is mentioned later in the same paragraph
LINE 146
Why did you use here 0,25 uM? Is not too low to see effects? might this be the reason for no apoptosis detected?
LINE 161
“After harvesting from xenograft mice, cells were maintained in co-culture with murine bone marrow stromal cells, simulating the protective bone marrow microenvironment.” I would move this sentence to the end of this paragraph
LINE 168 and forward
Please use genes in italics
LINE 167
You don’t use the genetic information at all in your interpretation of the results. Then, I think you should remove these part of the manuscript, both in methodology and in results sections
LINE 174
“as low as 1.0 uM...”
LINE 205
From now on you only work with these two cell lines. Maybe you should specify why.
LINE 230
Did you test cell viability too? did you detect synergy? In discussion section you mention both cell viability and metabolic activity (line 348), but in results section it is not clear, please show the results for synergy with cell viability.
LINE 257
There are many sentences where commas would help readers, for example here, line 305, line 311, etc. Please check through the whole manuscript.
LINE 269
I don't think you can say that. Differences are barely seen.
LINE 290
Please check again if this ref is stating what you mention. Moreover, this sentence and the following one should go to broaden information about previous studies in the Introduction section. The same is true for the sentences from line 296 to 301.
LINE 316
Do you have an explanation for REH cell line behavior?
LINE 321
“...MK-2206 induced...”
LINE 342
Why do you think that primary samples had apoptosis and cell lines didn’t? possible explanations? In primary samples 0,25 uM was not tested, but if we try to infer apoptosis levels from the data, it could be that the apoptosis levels were very low. So, maybe it is not so different after all.
LINE 342
Maybe MK-226 is acting as a cytostatic agent rather than cytotoxic at the low concentrations used in the present study (0,25 uM)
LINE 385 and forward
There are some mistakes with the use of subscripts and superscripts. Please check all of them throughout the text.
LINE 398
“...cells from primary samples...”
LINE 399
How many mice were used?
LINE 412
Bone marrow or spleen cells?
LINE 436
I think it would be more accurate if you say cell viability rather than proliferation. Please change through the whole manuscript.
LINE 370
Please, use the results with a lot of caution. It is critical to remember that short-term assays do not directly measure viability. Thus, if reproductive integrity is an important endpoint, confirmatory clonogenic assays should be considered.
Please read https://www.ncbi.nlm.nih.gov/pmc/articles/PMC6349361/
This is a limitation of the study and it should be mentioned
ABSTRACT
Please change accordingly to what has been mentioned so far.
Author Response
We thank reviewer 1 for the careful revision of our manuscript as well as the many suggestions for improvement. We have now revised the paper and think that it greatly benefits from the corrections made by reviewer 1. All changes are documented in color in the revised version of the paper. We hope that the manuscript is now suitable for publication in IJMS.
LINE 2
Reviewer 1: I believe the title is too ambitious, the results should be taken with a lot of caution
Response: We understand the point made by reviewer 1 and agree to change the title to a more neutral version: “Combined application of pan-AKT inhibitor MK-2206 and BCL-2 antagonist venetoclax in B-cell precursor acute lymphoblastic leukemia”.
LINE 45
Reviewer 1: Specify more clearly the involvement of PI3K/AKT in leukemia
Response: We apologize for the insufficient introduction into the PI3K/AKT pathway-mediated leukemogenesis and have therefore added the following paragraph for clarification: “Typical PI3K/AKT pathway alterations leading to leukemogenesis are mutations in tyrosine kinase genes (KIT, FMS, FLT3) or downstream signaling members like PI3K isoenzymes, PTEN or Ras oncoproteins [9]. Changes in protein expression and activity of PTEN, CK2 and AKT further result in increased tumor cell proliferation [10]. Aberrant PI3K/AKT signaling ultimately leads to uncontrolled cell growth and blockade of apoptotic cascades via several downstream proteins like GSK3β or 4EBP1 [9].“
LINE 47
Reviewer 1: In B-ALL
Response: We have included this term in the sentence: ”It further mediates resistance towards chemotherapeutic agents like glucocorticoids and poor prognosis in B‑ALL [8].“
LINE 51
Reviewer 1: In B-ALL
Response: We have now included this term in the sentence: “Regarding B‑ALL several PI3K/AKT pathways inhibitors have been designed and evaluated both preclinically and clinically with most studies focusing on PI3K kinase and mTOR complexes [10].“
LINE 52
Reviewer 1: Remove kinase
Response: We apologize for this duplication and removed the word “kinase”: “Regarding B‑ALL several PI3K/AKT pathways inhibitors have been designed and evaluated both preclinically and clinically with most studies focusing on PI3K and mTOR complexes [10].”
LINE 56
Reviewer 1: Explain why not to study GSK690693
Response: The main objective of the manuscript was to study the effects of MK‑2206 on B‑ALL cells. Of course the evaluation of a different AKT inhibitor, GSK690693, would be just as interesting but beyond the scope of the present research. We ultimately settled on MK-2206 because it seemed more specific compared to GSK690693 which is known to also inhibit AGC kinase family members to a certain extend.
LINE 55
Reviewer 1: and...? which parameters did they study? what results did they obtain?
see line 68
Response: As the results of the existing studies are discussed in the discussion section in great detail we only inserted a short sentence describing the results of the studies conducted in B-ALL cells in this part of the manuscript: “Levy et al. demonstrated that GSK690693 acted anti-proliferative and induced apoptosis [13,14] while three other manuscripts investigated pan‑AKT inhibitor MK‑2206 [15–17]. Both drugs induced significant anti-proliferative effects in leukemic cell lines and changes in AKT as well as downstream protein phosphorylation.”
LINE 68
Reviewer 1: Sparse means there are some articles. Which ones? Please describe. See comment on line 290
What's new in your study compared to the previous ones?
Response: As stated in the paragraph above there are preclinical studies by Simioni et al., Han et al. and Naderali et al. investigating the effect of MK-2206 in B-ALL. As correctly marked by reviewer 1 we have so far missed out on describing the results of these studies in a certain detail. We have now inserted a respective sentence explaining the results of these studies (see above). The novelty of our work is the investigation of numerous time points and concentrations tested, the cultivation of primary samples in a protective microenvironment as well as the combination of MK-2206 with venetoclax. This is expressed by the last paragraph of the introduction section, “We therefore aim to characterize the effect of MK‑2206 alone as well as in combination with BCL‑2 inhibitor venetoclax on B‑ALL cell lines and primary samples in a protective co‑culture environment. The investigation of this previously untested combination might offer insights into a potentially synergistic mechanism for acute leukemia abrogation.”
LINE 70
Reviewer 1: Last paragraph of introduction should refer to the aims of the study and not a summary of results, which are described in the abstract.
Response: The reviewer is absolutely correct and we changed the last paragraph of the introduction section: “We therefore aim to characterize the effect of MK‑2206 alone as well as in combination with BCL‑2 inhibitor venetoclax on B‑ALL cell lines and primary samples in a protective co‑culture environment. The investigation of this previously untested combination might offer insights into a potentially synergistic mechanism for acute leukemia abrogation.”
LINE 71
Reviewer 1: From the introduction information, there is no reason to use venetoclax, please explain, at least slightly
Response: This comment by reviewer 1 is a major shortage of our manuscript and we apologize for this lack of clarity. As requested we added an explaining sentence in the introduction section: “Leukemic cells often lack the inhibition of intrinsic apoptosis. Increased anti-apoptotic BCL‑2 signaling is frequently observed in ALL cells as well as further hematological neoplasms [31]. BCL‑2 downstream target and pathway member BAD directly interacts with AKT during apoptosis induction [32], justifying mutual targeting of AKT and BCL‑2.“
FIGURE 1
Reviewer 1: If there are two biological replicates, fig 1b could be the mean of the two replicates and include standard deviations?
Response: Thank you for this advice. It is true that displaying means and standard deviations of multiple replicates would increase the scientific meaningfulness of the data. However, as often with immunoblot data, standard deviations can be quite high, making it difficult to track the longitudinal changes of the separate replicates. This is the reason why we initially decided to plot the quantification results of a single experiment per cell line. After considering the advice made by both reviewers to include both replicates we have now plotted the individual data points of all immunoblots. The only exception is the REH cell line with very low basal pAKT levels. The depicted immunoblot was just above our standard requirements for quantification while the signal to background ratio in the second replicate was even lower, making it impossible to obtain a trustworthy quantification of the results. This issue is now described in the figure legend: “(b) Phospho-AKT and total AKT band intensities of two independent biological replicates including the blots shown in figure 1a were determined using Image Studio Lite 5.2 software. AKT phosphorylation was calculated as the ratio of phospho‑AKT/total AKT and normalized to the DMSO control separately for every time point. One REH replicate was not quantified due to the very low basal pAKT expression and therefore low signal to noise ratio not acceptable for reliable quantification.”
LINE 84
Reviewer 1: If you are talking about all cell lines, you should say 0,5 uM
Response: Of course this should be 0.5 µM, reviewer 1 is absolutely right. We have changed the passage accordingly: “Incubation with MK‑2206 induced an initial decrease in AKT activity in all cell lines which was detectable after 30 min incubation already, even at concentrations as low as 0.5 µM.”
LINE 85
Reviewer 1: I am afraid not always. I think you cannot state this.
Response: We do believe that the dephosphorylation of AKT is a concentration-dependent process but do agree with reviewer 1 that it is inappropriate to state this with an absolute term like we have used it in the manuscript. We therefore modified the sentence as follows: “Although the highest concentration (5 µM) resulted in the strongest dephosphorylation lower concentrations induced comparable effects.”
LINE 88
Reviewer 1: Please comment on REH cells
Response: Interpretation of changes in REH cells are hard to make due to the low basal expression. We have now stated this in the results section: “Due to the very low basal pAKT expression changes in REH are difficult to observe and interpret but suggest a decrease in AKT phosphorylation.”
FIGURE S1
Reviewer 1: It would be interesting to see the plots with the band intensities values, as in Fig1b
Response: It is of course correct that quantified band intensities would be useful for the interpretation of the MK-2006-induced pathway regulation. However, for the antigens and samples presented in those blots, this is quite difficult. In RS4;11 and REH cells the basal 4EBP1 and ERK1/2 phosphorylation is very low, resulting in very low signal-to-background ratios. Small irregularities within the background region used for normalization will therefore have a huge impact on the quantification result. The obtained intensity data would not be reliable at all. We thus decided to not quantify the influence of MK-2206 on 4EBP1 and ERK1/2 phosphorylation and instead rely on the description of the visual impression.
LINE 111
Reviewer 1: Shouldn't this sentence go to the previous paragraph? “REH cells demonstrated a very low overall pAKT 111 expression while total protein levels were comparable to the other cell lines.”
Response: As requested by reviewer 1 we have already described the effects of MK-2206 on REH cells in the previous paragraph were it fits in well (see above). We therefore modified the section as follows: “Similar to MK‑2206-induced AKT dephosphorylation PI3K/AKT signaling regulation was rather low in REH cells.”
LINE 120
Reviewer 1: Why did you chose these time points? did you test other time points? NALM-6 cell line at 48h recovers initial activity
Response: Observation time points of 48 h and 72 h are routinely tested for proliferation and metabolism-based assays in our as well as multiple other laboratories worldwide. Testing at earlier time points might imply that targeted signaling cascades are not fully translated into a certain biological effect (like proliferation or apoptosis). Checking at later time points can result, as indirectly pointed out by reviewer 1, in false negative results because the cells recover from therapy or induce evading mechanisms, ultimately leading to re-induction of proliferation, metabolism or blockade of apoptosis. The analysis of those really interesting questions, however, was behind the scope of this paper.
LINE 122
Reviewer 1: “0.05 μM MK-2206” I think this could be deleted, since it is mentioned later in the same paragraph
Response: Reviewer 1 is correct, we deleted the respective sentence.
LINE 146
Reviewer 1: Why did you use here 0,25 uM? Is not too low to see effects? might this be the reason for no apoptosis detected?
Response: It is true that the increase of the concentration we used for apoptosis studies might result in the detection of apoptotic cells. However, we decided to conduct our experiments with the concentration of 0.25 µM because this concentration induced a reduction of proliferation and tumor cell metabolism of ca. 20-30%. We deliberately wanted to explore the direct effects induced by targeted AKT inhibition instead of risking to detect the results of cytotoxic or cytostatic apoptotic off-target effects frequently observed in higher concentrations. As demonstrated in our co-culture experiments concentrations as low as 0.05 µM can be sufficient to induce apoptotic events.
LINE 161
Reviewer 1: “After harvesting from xenograft mice, cells were maintained in co-culture with murine bone marrow stromal cells, simulating the protective bone marrow microenvironment.” I would move this sentence to the end of this paragraph
Response: In this point we disagree with reviewer 1. We believe that it is important to very briefly explain the differences between the so far used cell culture method and the now described primary cell co-culture experiments. A major pitfall of co-culture studies is the fact that inhibitor experiments are often performed right after the initiation of co-culture which is inappropriate because the cells usually need some time to adapt and start proliferating. In our studies we cultivated the primary cells in co-culture and only commenced inhibitor testing once stable proliferation was achieved. Especially because the methods section is located at the end of the paper we strongly favor to briefly introduce our co-culture system before presenting the results.
LINE 168 and forward
Reviewer 1: Please use genes in italics
Response: This is of course correct and we apologize for the lack in correctness. We changed this throughout the manuscript.
LINE 167
Reviewer 1: You don’t use the genetic information at all in your interpretation of the results. Then, I think you should remove these part of the manuscript, both in methodology and in results sections
Response: The characterization of the mutational landscape of primary samples used in inhibitor studies is a standard approach in our and other laboratories. To put more emphasize on these interesting and important data we now inserted a respective paragraph in the discussion section: “The presence of a PIK3CA mutation as well as other pathogenic hotspot variants in patient #0159 did not interfere with the induction of MK‑2206-induced effects as well. The fact that the inhibitor produced significant anti-leukemic effects in primary samples harboring KMT2A rearrangements associated with a very grim prognosis further underlines its potential.”
LINE 174
Reviewer 1: “as low as 1.0 uM...”
Thanks for this advice. We now clarified the sentence as follows: “Interestingly, cell vitality dropped independently of pAKT expression with significant reductions starting at 1.0 µM and 0.5 µM for patients #0122 and #0159, respectively.”
LINE 205
Reviewer 1: From now on you only work with these two cell lines. Maybe you should specify why.
Response: Thank you for this advice. We have inserted a respective explanation: “Cell lines SEM and RS4;11 were selected based on their rather high basal pAKT expression as well as their molecular subtype, sharing the KMT2A rearrangement with the tested primary samples.”
LINE 230
Reviewer 1: Did you test cell viability too? did you detect synergy? In discussion section you mention both cell viability and metabolic activity (line 348), but in results section it is not clear, please show the results for synergy with cell viability.
Response: We apologize if the conducted experiments and interpretation are not clear in this case. We first assessed both proliferation and metabolic activity in SEM and RS4;11 cells when MK-2206 and venetoclax were applied simultaneously (Figure 5c,d). Accordingly, synergism was calculated for both, proliferation and metabolic activity. The values for proliferation are now inserted in a separate sentence: “Mild synergistic anti-proliferative responses were present in both cell lines (SEM: 0.0452; RS4;11: 0.0820).” The results of the metabolic activity are displayed in Figure 5f together with the values of the sequential application. As there were no major differences in the absolute results between proliferation and metabolic activity in the simultaneous application we subsequently focused on metabolic activity for the assays investigating sequential application regimens (Figure 5e).
LINE 257
Reviewer 1: There are many sentences where commas would help readers, for example here, line 305, line 311, etc. Please check through the whole manuscript.
Response: We apologize for the lack of conciseness and have revised the manuscript accordingly.
LINE 269
Reviewer 1: I don't think you can say that. Differences are barely seen.
Response: We agree with reviewer 1 that our claim might be a little bit too optimistic. We therefore modified the respective sentence as follows: “Similar to SEM cells, combined application resulted in a minor rise of Bax/mitochondria overlap.”
LINE 290
Reviewer 1: Please check again if this ref is stating what you mention. Moreover, this sentence and the following one should go to broaden information about previous studies in the Introduction section. The same is true for the sentences from line 296 to 301.
Response: We apologize for our incorrectness with this reference. The cited review describes the mechanisms of PI3K/AKT-related oncogenesis and we therefore corrected the sentence accordingly: “Dysregulated PI3K/AKT signaling is observed in many cancer entities including hematologic malignancies, suggesting AKT as a possible target for anti-leukemic targeting [32].” We have also added more explanatory information in the introduction section as requested by reviewer 1 (see above).
LINE 316
Reviewer 1: Do you have an explanation for REH cell line behavior?
Response: In this part of the discussion section we interpret the dephosphorylation of AKT and downstream signaling molecules 4EBP1 and ERK1/2. As REH cells exhibit a very low basal AKT, 4EBP1 and ERK1/2 phosphorylation there were no considerable changes observed. We therefore focused on the other three cell lines in this part of the discussion.
LINE 321
Reviewer 1:“...MK-2206 induced...”
Response: We changed the grammar of this sentence to increase clarity and conciseness as follows: “This phenomenon suggests that MK‑2206 induced off-target effects by binding a structurally related molecule so far unidentified.”
LINE 342
Reviewer 1: Why do you think that primary samples had apoptosis and cell lines didn’t? possible explanations? In primary samples 0,25 uM was not tested, but if we try to infer apoptosis levels from the data, it could be that the apoptosis levels were very low. So, maybe it is not so different after all.
Response: This question raised by reviewer 1 is very interesting and definitely worth discussing. We have therefore included possible explanations for the observed discrepancy: “Interestingly, and in contrast to the evaluated primary samples MK‑2206 did not induce apoptosis in any of the observed cell lines. This might be due to the technical fact that primary cultures are more fragile than robust and immortal cell culture systems. Otherwise, the molecular background of the patient samples is different from the cell lines.” Also, reviewer 1 is correct in mentioning the different concentrations used for apoptosis studies. Apoptosis levels were indeed rather low within the concentration range tested in the cell lines; however, patient #0159 exhibited elevated cell death levels even at a concentration as low as 0.05 µM while no apoptosis was detected at 0.25 µM in cell lines. Using a different method (immunoblot of cleaved caspase 3) no signs of apoptosis were observed in cell lines with concentrations up to 5 µM.
LINE 342
Reviewer 1: Maybe MK-226 is acting as a cytostatic agent rather than cytotoxic at the low concentrations used in the present study (0,25 uM)
Response: We assume that reviewer 1 refers to the term “cytotoxic” in line 352 instead of 342 of the original manuscript. In this case and after re-checking our raw data we agree that reviewer 1 is right to claim that the induced effect is rather cytostatic than cytotoxic. We therefore corrected the sentence: “To check whether the increased cytostatic effect was due to raised apoptosis signaling we performed functional apoptosis analysis on a molecular level.”
LINE 385 and forward
Reviewer 1: There are some mistakes with the use of subscripts and superscripts. Please check all of them throughout the text.
Response: This claim is absolutely true and we apologize for that. There must have been an error during copy-pasting the text from our original draft to the IJMS template. We have corrected the respective passages accordingly.
LINE 398
Reviewer 1: “...cells from primary samples...”
Response: We have added the term to increase conciseness and clarity: “For patient-derived xenograft (PDX) model generation 2.5 x 106 cells per primary sample were orthotopically xenografted into 8-12 week old male or female NOD scid gamma mice (NOD.Cg-Prkdcscid Il2rgtm1Wjl/SzJ, NSG, Charles River Laboratories, Sulzfeld, Germany).”
LINE 399
Reviewer 1: How many mice were used?
Response: The respective primary samples were injected into one mouse each and after tumor cell isolation the blasts were re-injected into the next PDX generation. Cells derived of PDX generation 3 were used for the in vitro co-culture experiments so a total of six mice were directly involved in this experiment. We added this information in the methods section: “For patient-derived xenograft (PDX) model generation 2.5 x 106 cells per primary sample were orthotopically xenografted into 8-12 week old male or female NOD scid gamma mice (NOD.Cg-Prkcdscid Il2rgtm1Wjl/SzJ, NSG, Charles River Laboratories, Sulzfeld, Germany).” … “Tumor cells were then serially transplanted into the next PDX generation until generation 3.” … “PDX-derived primary tumor cells isolated from spleens of the third xenograft generation were subsequently cultured in vitro using murine bone marrow stroma cell line OP‑9 as feeder layer.”
LINE 412
Reviewer 1: Bone marrow or spleen cells?
Response: Please excuse this missing important information. We have included it in the methods section: “PDX-derived primary tumor cells isolated from spleens of the third xenograft generation were subsequently cultured in vitro using murine bone marrow stroma cell line OP‑9 as feeder layer.”
LINE 436
Reviewer 1: I think it would be more accurate if you say cell viability rather than proliferation. Please change through the whole manuscript.
Response: In this point we disagree with reviewer 1. In our methodology we deliberately focused on tumor cell proliferation as a marker for growth propagation. Trypan blue staining and subsequent counting of viable cells as performed in our experiments is not suitable to determine cell viability because the number of dead cells is not assessed. Therefore, the viability of the sample cannot be determined. To get an idea of the the cells’ activity we used the WST-1 assay to measure the cells’ metabolism. This value usually correlates well with the viability of a sample.
LINE 370
Reviewer 1: Please, use the results with a lot of caution. It is critical to remember that short-term assays do not directly measure viability. Thus, if reproductive integrity is an important endpoint, confirmatory clonogenic assays should be considered.
Please read https://www.ncbi.nlm.nih.gov/pmc/articles/PMC6349361/
This is a limitation of the study and it should be mentioned
Response: Reviewer 1 definitely has a point with this important remark. Short-term in vitro assays are not suitable to draw conclusions on clinical efficacy or persistent anti-tumor effects. We have already mentioned this point in the discussion section when feedback loops and evading mechanisms were considered. Also, it was not the scope of this manuscript to make such claims. We rather focused on identifying the underlying molecular and biochemical pathways of the MK-2206-induced effects. Long-term and in vivo studies are necessary to further evaluate if the regulation of those mechanisms is sufficient to reduce the tumor burden in a clinical setting. To remind of these points we changed the concluding section of the discussion to a more cautious version: “In summary this study investigates the short-term mechanistic effects of MK‑2206 incubation, demonstrating that, MK‑2206 application is a powerful tool to induce a significant decrease in leukemia proliferation throughout observation periods up to 72 h. This was true even in primary blasts cultivated in a protective microenvironment. It underlines the anti-leukemic potential of AKT inhibition and evaluates the synergistic combined application with venetoclax previously untested in ALL. These results raise hopes for further long-term observation and in vivo evaluation of the AKT inhibitor alone or in combination with other small molecules.” We further thank reviewer 1 for the interesting literature on clonogenic assays. Although highly relevant, this type of assay is not suitable for our cell types as leukemic blasts do not grow in monolayers or form colonies.
ABSTRACT
Reviewer 1: Please change accordingly to what has been mentioned so far.
We have made minor changes in the abstract, mainly clarifying which analyses have been performed on which cells and models. Please refer to the revised manuscript for red-marked track changes.
Reviewer 2 Report
In this research manuscript, Richter and colleagues investigate the effect of AKT inhibitor MK-2206 alone and in combination with venetoclax on B-ALL cell lines and in a coculture model of primary cells obtained from patients with B-ALL. The study shows data on the biochemical level (protein phosphorylation) and the cellular level (assessment of proliferation, metabolic activity and apoptosis induction). The study is overall well planned and the data show a clear picture. The manuscript is well written and easy to follow.
Before publication, I recommend that the authors address the following points:
- In line 101/102 of the manuscript, it is written that “AKT downstream targets 4EBP1 and GSK3β were analyzed by immunoblot and indicated increased 4EBP1 and GSK3β activity after MK-2206 treatment.” I assume it should mean decreased
- In Figure 1b, the quantification of the immunoblot data should not only show the pictured experiment, but rather the mean of all experiments. Ideally a third replicate should be made and then mean plus/minus sd of all replicates shown.
- In Figure 3, sub-figures are indicated by upper-case letters, whereas in all other figures lower-case letters are used.
- Beginning with Figure 3, only data from two of the four cell lines is shown. I assume this is because the other two lines have low basal pAKT levels? This should either be clarified in the text, or, alternatively, data should be shown for all cell lines.
- The lack of apoptosis induction by MK-2206 on B-ALL cell lines (shown in Figure 3b-c) could be due to the rather low dose of 0.25 µM. I recommend to repeat the experiment using higher doses (up to 1 µM) and see if apoptosis induction occurs.
- Section 2.3 would benefit from an introductory sentence explaining the experimental setup using a xenograft derived model.
- It would be of interest to analyze the effects of MK-2206 in the PDX-coculture model using a patient with low or absent pAKT levels. If MK-2206 effects are still present in this case, it would hint towards additional unknown targets of MK-2206 (as already implied in the discussion).
- In Figure 5, the PBMC viability assay should be repeated using an incubation period of 72 h in order to have the same duration of drug exposure as in the cell line experiments.
- The discussion should include a paragraph on the clinical implications of the data. Given that B-ALL is an aggressive disease that requires high-dose chemotherapy, the use of a novel antiproliferative (but not antiapoptotic) agent should be discussed.
Author Response
We thank reviewer 2 for the careful revision of our manuscript as well as the suggestions for improvement. We have now revised the paper and think that it greatly benefits from the corrections made by reviewer 2. All changes are documented in color in the revised version of the paper. We hope that the manuscript is now suitable for publication in IJMS.
- In line 101/102 of the manuscript, it is written that “AKT downstream targets 4EBP1 and GSK3β were analyzed by immunoblot and indicated increased 4EBP1 and GSK3β activity after MK-2206 treatment.” I assume it should mean decreased
Response: In this special case and in contrast to AKT protein activity the dephosphorylation of 4EBP1 results in protein function activation. We have clarified this confusing fact: “Protein expression and phosphorylation of AKT downstream target 4EBP1 was analyzed by immunoblot and demonstrated reduced protein phosphorylation, thus indicating increased 4EBP1 activity after MK‑2206 treatment.”
- In Figure 1b, the quantification of the immunoblot data should not only show the pictured experiment, but rather the mean of all experiments. Ideally a third replicate should be made and then mean plus/minus sd of all replicates shown.
Response: This important issue was also mentioned by reviewer 1. It is true that displaying means and standard deviations of multiple replicates would increase the scientific meaningfulness of the data. However, as often with immunoblot data, standard deviations can be quite high, making it difficult to track the longitudinal changes of the separate replicates. This is the reason why we initially decided to plot the quantification results of a single experiment per cell line. After considering the advice made by both reviewers to include both replicates we have now plotted the individual data points of all immunoblots. The only exception is the REH cell line with very low basal pAKT levels. The depicted immunoblot was just above our standard requirements for quantification while the signal to background ratio in the second replicate was even lower, making it impossible to obtain a trustworthy quantification of the results. This issue is now described in the figure legend: “(b) Phospho-AKT and total AKT band intensities of two independent biological replicates including the blots shown in figure 1a were determined using Image Studio Lite 5.2 software. AKT phosphorylation was calculated as the ratio of phospho‑AKT/total AKT and normalized to the DMSO control separately for every time point. One REH replicate was not quantified due to the very low basal pAKT expression and therefore low signal to noise ratio not acceptable for reliable quantification.”
- In Figure 3, sub-figures are indicated by upper-case letters, whereas in all other figures lower-case letters are used.
Response: Thank you very much for this advice. The error is now corrected and we also inserted arrows indicating the morphological changes in figure 3a.
- Beginning with Figure 3, only data from two of the four cell lines is shown. I assume this is because the other two lines have low basal pAKT levels? This should either be clarified in the text, or, alternatively, data should be shown for all cell lines.
Response: We apologize for this insufficient explanation. We have included a respective statement at the beginning of the combination experiments conducted in SEM and RS4;11 cells: “Cell lines SEM and RS4;11 were selected based on their rather high basal pAKT expression as well as their molecular subtype, sharing the KMT2A rearrangement with the tested primary samples.”
- The lack of apoptosis induction by MK-2206 on B-ALL cell lines (shown in Figure 3b-c) could be due to the rather low dose of 0.25 µM. I recommend to repeat the experiment using higher doses (up to 1 µM) and see if apoptosis induction occurs.
Response: It is true that the increase of the concentration we used for apoptosis studies might result in the detection of apoptotic cells. However, we decided to conduct our experiments with the concentration of 0.25 µM because this concentration induced a reduction of proliferation and tumor cell metabolism of ca. 20-30%. We deliberately wanted to explore the direct effects induced by targeted AKT inhibition instead of risking to detect the results of cytotoxic or cytostatic apoptotic off-target effects frequently observed in higher concentrations. As demonstrated in our co-culture experiments concentrations as low as 0.05 µM can be sufficient to induce apoptotic events.
- Section 2.3 would benefit from an introductory sentence explaining the experimental setup using a xenograft derived model.
Response: Thank you for this advice, we think this is a great idea to increase the clarity and understandability of the model and results. We incorporated a respective section: “Primary cells were orthotopically xenografted into immunocompromised mice and tumor cell engraftment was regularly checked by peripheral blood flow cytometry. When blast frequencies reached 30% mice were sacrificed and tumor cells were isolated from bone marrow and spleen.”
- It would be of interest to analyze the effects of MK-2206 in the PDX-coculture model using a patient with low or absent pAKT levels. If MK-2206 effects are still present in this case, it would hint towards additional unknown targets of MK-2206 (as already implied in the discussion).
Response: This is indeed a very interesting question. Unfortunately, our ALL biobank currently comprising six MLL rearranged adult B-ALL samples does not include a patient with absent AKT phosphorylation. With patients #0122 and #0159 we already selected those samples with the highest and lowest basal AKT activity from the available group.
- In Figure 5, the PBMC viability assay should be repeated using an incubation period of 72 h in order to have the same duration of drug exposure as in the cell line experiments.
Response: In this point we agree with reviewer 2 that it would be interesting to assess the effect of the inhibitors on healthy PBMCs after 72 h incubation. However, this assay is set to detect acute cytotoxic events induced by MK-2206 or venetoclax and those would be definitely seen after 24 h incubation. Further, primary blood cell viability decreases with ex vivo culture time and longer incubation periods might therefore result in false-low viability values. Still, we do believe that this issue is worth noting in the discussion section and thus modified the respective sentence: “As this combination is tested in acute leukemia cell lines for the first time and was not evaluated regarding cytotoxic effects on healthy blood cells, we first demonstrated that the application was safe and not affecting erythrocytes and PBMCs after short-term incubation.”
- The discussion should include a paragraph on the clinical implications of the data. Given that B-ALL is an aggressive disease that requires high-dose chemotherapy, the use of a novel antiproliferative (but not antiapoptotic) agent should be discussed.
Response: In our opinion this point raised by reviewer 2 is a little bit too early at this stage of the current research. It is of course true that B-ALL therapy urgently needs novel drugs. The development and clinical use of novel therapeutic agents and strategies, however, is a very long process. While our study only focused on mechanistic biochemical and cell biological pathways of MK-2206 and venetoclax efficacy, many more experiments including in vivo testing of pharmacokinetics, efficacy as well as tolerability are necessary to draw any slight conclusion about clinical implications. This is well beyond the aim of this current research.
Round 2
Reviewer 2 Report
My comments and concerns have been sufficiently addressed. I have no further comments and therefore recommend to accept the manuscript for publication.